# Association of C-peptide and lipoprotein(a) as two predictors with cardiometabolic biomarkers in patients with type 2 diabetes in KERCADR population-based study

Mohammad Reza Mahmoodi[¤a]*, Hamid Najafipour[¤b]

Physiology Research Center, Institute of Neuropharmacology, Kerman University of Medical Sciences, Kerman, Iran

¤a Current address: Department of Nutrition, Faculty of Public Health, Kerman University of Medical Sciences, Kerman, Iran
¤b Current address: Department of Physiology and Pharmacology, Faculty of Medicine, Kerman University of Medical Sciences, Kerman, Iran
* mahmoodimr@yahoo.com

**Data Availability Statement:** The data that support the findings of this study are available without any restriction in the Supporting Information files.

## Abstract

We sought association between serum Lipoprotein(a) and C-Peptide levels as two predictors with cardiometabolic biomarkers in patients with type 2 diabetes mellitus. This nested case-control study was conducted on 253 participants with type 2 diabetes mellitus and control from the second phase of the KERCADR cohort study. The participants were randomly allocated into case and control groups. The quantitative levels of Lipoprotein(a) and C-Peptide were measured by ELISA. Atherogenic indices of plasma were measured. The plasma Atherogenic Index of Plasma significantly decreased ($P = 0.002$) in case-male participants, and plasma Castelli Risk Index II level significantly increased ($P = 0.008$) in control-male participants with the highest dichotomy of Lipoprotein(a). The plasma Atherogenic Index of Plasma level in case-female participants significantly increased ($P = 0.023$) with the highest dichotomy of C-Peptide. Serum C-Peptide level significantly increased ($P = 0.010$ and $P = 0.002$, respectively) in control-male participants with the highest dichotomies of Atherogenic Index of Plasma and Castelli Risk Index I. There was a significant association between the highest quartile of C-Peptide and higher anthropometric values in case participants; and higher atherogenic indices of plasma and anthropometric values in control participants. Raised serum C-peptide than raised Lipoprotein(a) can be a prior predictor for cardiometabolic disease risk in healthy participants and patients with type 2 diabetes mellitus with increased cardiometabolic biomarkers. Case and control males with general and visceral obesity and case and control females with visceral obesity are exposure to increased C-peptide, respectively. Lipoprotein(a) may be risk independent biomarker for type 2 diabetes mellitus. Reducing raised Lipoprotein(a) levels to less than 30ng/ml with strict control of low density lipoprotein cholesterol would be the best approach to prevent coronary artery disease consequences. It is suggested that a screening system be set up to measure the Lp(a) levels in the community for seemingly healthy people or individuals with one or more cardiometabolic biomarkers.

**Funding:** The Vice chancellor for Research of Kerman University of Medical Sciences had no role in study design, data collection and analysis, decision to publish, or preparation of the manuscript. The grant number of this research was (Grant ID: 95000643), however, the Ethics Certificate code was (IR.KMU.REC. 1395.938).

**Competing interests:** No potential conflict of interest relevant to this article was reported.

## Introduction

Raised concentrations of serum C-peptide have been observed in patients with type 2 diabetes mellitus (T2DM) and insulin resistance [1]. C-peptide as a probable risk biomarker for coronary artery disease (CAD) displayed a critical role in the pathogenesis of atherosclerosis [2]. The researchers have demonstrated increased serum C-peptide, reflecting raised insulin secretion, is as a biomarker for obesity and insulin resistance, especially in T2DM and metabolic syndrome (MS) [3]. Raised serum C-peptide is associated with an augmented chance of CAD in patients with T2DM. Fasting C-peptide, together with traditional risk factors such as hypertension, obesity, and hyperlipidemia remarkably augment the risk of clinical CAD in patients with T2DM [4]. Therefore, a relationship was found between serum C-peptide and macrovascular complications such as dyslipidemia, hypertension, and CAD in patients with T2DM [5]. Nakamura and colleagues revealed that patients in the CAD group without T2DM than in the Non-CAD group have higher significant levels of postprandial plasma insulin and dyslipidemia and raised this biomarker might play a critical role in the progress of atherosclerosis even before diabetes occurrence [6]. Raised serum C-peptide as a predictor of premature coronary incidents than impaired glucose tolerance is correlated to the occurrence of myocardial infarction (MI) and CAD in the popular community [7]. Elevated C-peptide levels significantly increased the risk of cardiovascular mortality independently of recognized diabetes situation in patients with atherosclerotic cardiovascular disease [8]. Plasma triglyceride/high-density lipoprotein cholesterol (TG/HDL-C) ratio allocates an accessible method to recognize patients at advanced cardiometabolic risk within a populace of increased risk of T2DM, insulin resistance, and CAD [9].

The results of studies revealed that raised lipoprotein(a) (Lp(a)) is an independent predictable risk biomarker for the severity of new-onset CAD in patients with T2DM [10, 11]. Raised Lp(a) and LDL-cholesterol (LDL-C) as independent prognosticators of the intensity of CAD were remarkable in people with premature CAD [12]. Raised serum Lp(a) concentrations can influence the prognosis of cardiovascular events in patients with pre-diabetes [13]. Therefore, it indicated that raised serum Lp(a) concentration is associated with a worse prognosis of symptomatic CAD in patients with T2DM [14]. In patients with T2DM, raised serum Lp(a) is strongly associated with increment risk for cardiovascular events. These evaluations may lead to better recognition of T2DM patients with high levels of Lp(a) who might obtain advantages from Lp(a)-lowering therapeutic procedures [15]. The results indicated that serum Lp(a) levels in patients with T2DM are significantly higher. Serum Lp(a) levels have a positive correlation with total cholesterol (TC) and LDL-C in these patients [16]. Although raised serum Lp(a) is a strong predictor of premature CAD in patients with T2DM, among the individuals with premature CAD, patients with T2DM have lower Lp(a) levels than patients without T2DM [17]. In the clinical setting, the determination of serum Lp(a) for describing cardiovascular disease (CVD) risk should measure in asymptomatic patients with a family history of coronary heart disease and patients at intermediate or high CVD risk [18].

This study sought to evaluate C-peptide and Lp(a) levels in patients with and without T2DM. Additionally, to find out whether there is an association between serum C-peptide and Lp(a) levels as two predictors with biomarkers of cardiometabolic disease risk in patients with T2DM and healthy people based on genders in KERCADR study as an Iranian community.

## Materials and methods

### Participants eligibility and study design

The current research is a nested case-control study conducted on participants with T2DM and control from the second phase of the KERCADR population-based cohort study. Over 10000

individuals aged 15–75 years old were enrolled in this household cohort survey (second phase) by non-proportional to size one-stage cluster sampling on Kerman province residences. Trained endocrinologists, internal specialists, general practitioners, clinical psychologists and dentists have assessed the study participants by person-assisted questionnaires regarding different non-communicable disease risk factors. The study protocol, the enrollment of individuals, and the sampling method were explained in detail, previously [19, 20]. Written informed consent was signed by all of the participants after ensuring they are well understanding of the harm and benefits of participation in the survey. For participants under 18, informed consent was acquired from both themselves and their parents, and they usually attended the interview site accompanied by their parents.

The proportion between cases and controls became 1:1 due to the precise control of confounding variables in the study. Hence, the study groups did not vary regarding specified and impressive confounders. Therefore, a matched healthy control was carefully chosen for each case from among participants in the KERCADR cohort study. One hundred twenty-eight controls and 125 participants with T2DM were randomly selected from the KERCADR population-based study.

The eligibility criteria of control participants from both genders were 1) agreement to attend in the study and sign the informed consent, 2) Body mass index (BMI) lower than 30, 3) lack of history of high blood pressure, and 4) lack of history of T2DM, MI, stroke, cardiovascular disease, active cancer, liver, kidney, and thyroid dysfunction, and infectious diseases. The eligibility criteria of participants with T2DM from both genders were 1) agreement to attend in the study and sign the informed consent, 2) the diagnosis of T2DM at least for one year, 3) BMI lower than 30, 4) lack of history of high blood pressure, 5) lack of history of MI, stroke, cardiovascular disease, active cancer, liver, kidney, and thyroid dysfunction, and infectious diseases, 6) participants with T2DM receive either therapeutic diet or therapeutic diet with a combination of oral anti-glycemic drugs.

The review panel and ethics committee of the Vice-chancellor for Research of Kerman University of Medical Sciences approved the protocol (Approval ID: IR.KMU.REC. 1395.938).

## Clinical and biochemical examinations

**Cardiometabolic biomarkers.** As formerly explained in the other studies [19, 20], all biochemical measurements were fulfilled to conform to the standard procedure. At the beginning of the study, after 12–14 h fasting, blood samples were collected through cephalic venipuncture and transferred to EDTA tubes. Plasma samples after separation were stored at –80°C until a final assay for cardiometabolic biomarkers could fulfill. Cardiometabolic biomarkers composed of fasting blood sugar (FBS), glycosylated hemoglobin (HbA1c), TC, LDL-C, HDL-C, TG, systolic blood pressure (SBP), diastolic blood pressure (DBP), waist circumference (WC), hip circumference (HC), waist to hip ratio (WHR), weight, and BMI.

FBS (KIMIA Kit, Code 890410, Iran) was measured using the glucose oxidase-peroxidase technique. HbA1C (NYCOCARD Kit, Code 1042184, Austria) was quantified conforming to Bio-Rad Variant High-Performance Liquid Chromatography [HPLC] assay. TC (KIMIA Kit, Code 890303, Iran), HDL-C (PARS Kit, Code 89022, Iran), as well as LDL-C were calculated based on the Friedewald formula [LDL-C = TC–(HDL-C + TG/5)]. TG (KIMIA Kit, Code 890201, Iran) was measured by standard enzymatic technique [19, 20].

**Clinical and anthropometry assessment.** BP was recorded using an automated standard mercury manometer (Model RIESTER, Germany) after at least 10 min of rest in a chair and arm supported at heart level. Weight and BMI (the weight in kilograms divided by the square of the height in meters) of participants were measured and recorded in questionnaires. WC

was also measured using a non-stretchable measuring tape at the umbilical level without any pressure to the body surface.

**Atherogenic indices of plasma.** Atherogenic indices of plasma include the Atherogenic Index of Plasma (AIP), Castelli Risk Index I (CRI I), Castelli Risk Index II (CRI II), and Atherogenic Coefficient (AC). AIP or TG/high-density lipoprotein cholesterol (TG/HDL-C) ratio is a logarithmic transformation of the ratio of molar concentrations of TG to HDL-C. CRI I is the ratio of TC to HDL-C and CRI II is the ratio of LDL-C to HDL-C. AC is the ratio of non-HDL-C to HDL-C.

**Determination of serum lipoprotein (a).** The quantitative levels of Lp(a) were measured using an enzyme-linked immune sorbent assay (ELISA) kit (Lipoprotein(a), Hangzhou East-biopharm Co., China; Cat. No: CK-E10852) based on the biotin double antibody sandwich technology to assay the Human Lp(a). Specific antibody against (Lp(a)) antigen was coated in wells and our antigen was sandwiched between primary and secondary HRP-coated antibodies. After washing was completed, we added A and B substrate solutions. The substrate became blue color in wells that contained antibody-antigen-enzyme-antibody complex. The reaction was terminated by the addition of a stop solution, and the color change is measured at a wavelength of 450 nm by the ELISA reader (reference wavelength 630 nm). Finally, the concentration of Lp(a) (ng/ml) in the samples was determined by comparing the O.D. of the samples to the standard curve (Sensitivity: 0.23ng/ml).

**Determination of serum C-peptide.** The Human C-peptide levels in serum was measured by using an enzyme-linked immune sorbent assay (ELISA) kit (C-peptide, Monobind, USA; Catalog #: 2725–300). Specific antibody against each antigen (C-peptide) was coated in wells and our antigen was sandwiched between primary and secondary HRP-coated antibodies. Then, color progress within the 10 minutes was assessed at 450 nm by the ELISA reader instrument (ELM-2000).

## Statistical analysis

Statistical analysis was performed using IBM SPSS Statistics software, version 22.0. The normal distribution of biomarkers was examined by the Kolmogorov-Smirnov test. The P. value < 0.05 was assumed significance. We compared the mean differences of serum Lp(a) and C-peptide levels between the case and the control participants in each gender group by independent t-test (Table 1). Then, the levels of atherogenic indices of plasma dichotomized into AIP level ≤ 0.4347 and > 0.4347; CRI I and AC levels ≤ 4.2093 and > 4.2093; and plasma CRI II level ≤ 2.5156 and > 2.5156. Independent t-test analyzed the mean differences of the serum Lp(a) and C-peptide levels between the case and the control groups for median dichotomies of atherogenic indices of plasma (Table 2). Independent t-test analyzed the mean differences of the cardiometabolic biomarkers between lowest and highest quartiles Lp(a) and C-peptide for case and control groups (Table 3). The Lp(a) levels dichotomized into Lp(a)

**Table 1. Mean ± SE and Interquartile range* of serum lipoprotein (a) and C-Peptide‡ levels of participants according to gender.**

|  | Total Participants (n = 253) | | P value | Male (n = 137) | | P value | Female (n = 116) | | P value |
|---|---|---|---|---|---|---|---|---|---|
|  | Case (n = 125) | Control (n = 128) |  | Case (n = 70) | Control (n = 67) |  | Case (n = 55) | Control (n = 61) |  |
| Lipoprotein (a) | 68.04±3.74 (32.10) | 66.98±2.57 (29.06) | 0.815 | 64.45±4.40 (31.00) | 68.18±3.79 (31.33) | 0.523 | 72.61±6.37 (43.92) | 65.65±3.42 (25.35) | 0.339 |
| C-Peptide | 1.60±0.08 (1.00) | 1.42±0.06 (0.75) | 0.067 | 1.52±0.10 (0.90) | 1.45±0.08 (0.88) | 0.542 | 1.69±0.12 (1.10) | 1.39±0.08 (0.60) | 0.048 |

* Interquartile range is in bracket.

‡ Based on ng/ml.

**Table 2. Mean ± SE* serum lipoprotein (a) and C-Peptide levels‡ in lower and upper dichotomies of atherogenic indices of plasma¶ according to gender.**

| Biomarkers | AIP (Case-Male) | | | AIP (Case-Female) | | |
|---|---|---|---|---|---|---|
| | Lowest Dichotomy | Highest Dichotomy | *Sig.* | Lowest Dichotomy | Highest Dichotomy | *Sig.* |
| **Lipoprotein (a)** | 76.75 ± 8.28 | 55.76 ± 4.31 | **0.030** | 88.44 ± 11.51 | 64.91 ± 7.42 | 0.083 |
| **C-Peptide** | 1.40 ± 0.14 | 1.61 ± 0.13 | 0.283 | 1.40 ± 0.18 | 1.83 ± 0.15 | 0.101 |
| | AIP (Control-Male) | | | AIP (Control-Female) | | |
| **Lipoprotein (a)** | 74.12 ± 5.39 | 59.67 ± 5.01 | 0.058 | 68.38 ± 4.53 | 59.28 ± 4.10 | 0.187 |
| **C-Peptide** | 1.23 ± 0.08 | 1.65 ± 0.13 | **0.010** | 1.32 ± 0.10 | 1.57 ± 0.12 | 0.162 |
| | CRI I & AC (Case-Male) | | | CRI I & AC (Case-Female) | | |
| **Lipoprotein (a)** | 66.72 ± 5.58 | 62.18 ± 6.87 | 0.610 | 85.53 ± 10.22 | 61.02 ± 7.39 | 0.058 |
| **C-Peptide** | 1.46 ± 0.13 | 1.58 ± 0.14 | 0.512 | 1.54 ± 0.14 | 1.82 ± 0.19 | 0.248 |
| | CRI I & AC (Control-Male) | | | CRI I & AC (Control-Female) | | |
| **Lipoprotein (a)** | 63.38 ± 5.33 | 73.05 ± 5.48 | 0.212 | 63.79 ± 5.17 | 67.78 ± 4.41 | 0.566 |
| **C-Peptide** | 1.20 ± 0.09 | 1.69 ± 0.13 | **0.002** | 1.33 ± 0.13 | 1.47 ± 0.08 | 0.368 |
| | CRI II (Case-Male) | | | CRI II (Case-Female) | | |
| **Lipoprotein (a)** | 64.42 ± 5.52 | 66.51 ± 7.59 | 0.822 | 84.10 ± 10.32 | 63.47 ± 6.78 | 0.102 |
| **C-Peptide** | 1.52 ± 0.13 | 1.42 ± 0.14 | 0.579 | 1.69 ± 0.15 | 1.56 ± 0.13 | 0.555 |
| | CRI II (Control-Male) | | | CRI II (Control-Female) | | |
| **Lipoprotein (a)** | 65.00 ± 5.73 | 70.61 ± 5.09 | 0.468 | 63.64 ± 5.53 | 67.53 ± 4.21 | 0.574 |
| **C-Peptide** | 1.29 ± 0.12 | 1.57 ± 0.11 | 0.095 | 1.30 ± 0.14 | 1.49 ± 0.08 | 0.244 |

* Independent t-test analyzed the differences (Mean ± SE) lipoprotein (a) and C-Peptide levels that dichotomized based on median of atherogenic indices of plasma for case and control groups.

¶ Atherogenic Index of Plasma (AIP); Castelli Risk Index I (CRI I); Castelli Risk Index II (CRI II); Atherogenic Coefficient (AC).

‡ Based on ng/ml.

level $\leq$ 53.47 and > 53.48 in the cases; and $\leq$ 60.66 and > 60.66 in the controls. The levels of serum C-peptide dichotomized into serum C-peptide level $\leq$ 1.37 and > 1.37 in the case-male group; $\leq$ 1.44 and > 1.44 in the case-female group; $\leq$ 1.34 and > 1.34 in the control-male group; and $\leq$ 1.33 and > 1.33 in the control-female group. Independent t-test analyzed the mean differences in the atherogenic indices of plasma between the case and the control groups for median dichotomies of serum Lp(a) and C-peptide levels (Table 4). The Pearson correlation coefficient was calculated to examine the relationship between atherogenic indices and serum Lp(a) and C-peptide levels (Table 5). Independent t-test analyzed the mean differences in the plasma cardiometabolic biomarkers between the case and the control groups for median dichotomies of serum Lp(a) and C-peptide levels (Table 6). The Scatter Plot graph was also designed to examine the correlation between atherogenic indices of plasma and serum Lp(a) and C-peptide levels for both case and control groups according to gender (Figs 1 & 2).

## Results

The mean (±SE) and interquartile range of serum Lp(a) and C-peptide levels of participants were shown in case and control groups according to gender in Table 1. There was a significant difference among case and control female groups for C-peptide ($P$ = 0.048). The mean (±SD) age of participants was 47.79±5.65 years.

Results showed that the serum Lp(a) levels significantly decreased ($P$ = 0.030) only in the male case participants with the highest dichotomy of AIP. In general, the serum Lp(a) levels in the male and female cases and controls decreased with the highest dichotomy of AIP (Table 2). The level of serum C-Peptide significantly increased ($P$ = 0.010 and $P$ = 0.002, respectively)

**Table 3. Mean ± SE* of atherogenic indices¶ and cardiometabolic biomarkers‡ of plasma according to lowest and highest quartiles of lipoprotein (a) and C-Peptide levels.**

| Biomarkers | Lipoprotein (a) (Case) | | | Lipoprotein (a) (Control) | | |
|---|---|---|---|---|---|---|
| | Lowest Quartile (n = 52) | Highest Quartile (n = 30) | *Sig.* | Lowest Quartile (n = 38) | Highest Quartile (n = 35) | *Sig.* |
| **AIP** | 0.66 ± 0.04 | 0.42 ± 0.04 | **0.001** | 0.45 ± 0.05 | 0.37 ± 0.03 | 0.463 |
| **CRI I** | 4.59 ± 0.20 | 3.93 ± 0.18 | 0.146 | 4.16 ± 0.14 | 4.46 ± 0.14 | 0.546 |
| **CRI II** | 2.32 ± 0.12 | 2.34 ± 0.16 | 1.000 | 2.46 ± 0.10 | 2.94 ± 0.12 | **0.035** |
| **AC** | 3.59 ± 0.20 | 2.93 ± 0.18 | 0.146 | 3.16 ± 0.14 | 3.46 ± 0.14 | 0.546 |
| **Age** | 48.17 ± 0.74 | 47.53 ± 0.86 | 0.956 | 49.11 ± 0.96 | 47.40 ± 0.99 | 0.591 |
| **Weight (Kg)** | 72.53 ± 1.36 | 69.77 ± 1.95 | 0.669 | 67.74 ± 1.76 | 68.60 ± 1.89 | 0.985 |
| BMI (Kg/m$^2$) | 26.78 ± 0.31 | 26.08 ± 0.54 | 0.670 | 25.19 ± 0.57 | 24.68 ± 0.60 | 0.187 |
| **WC (cm)** | 91.73 ± 1.21 | 89.90 ± 1.79 | 0.851 | 87.63 ± 1.68 | 90.09 ± 1.57 | 0.741 |
| **WHR** | 0.92 ± 0.01 | 0.92 ± 0.01 | 0.991 | 0.89 ± 0.01 | 0.91 ± 0.01 | 0.623 |
| **SBP (mmHg)** | 112.6 ± 1.8 | 113.1 ± 2.6 | 0.998 | 115.0 ± 2.1 | 106.3 ± 2.2 | **0.019** |
| **DBP (mmHg)** | 75.4 ± 1.2 | 71.6 ± 1.7 | 0.206 | 74.9 ± 1.4 | 69.7 ± 2.0 | 0.147 |
| **FBG (mg/dl)** | 168.6 ± 7.6 | 181.7 ± 12.8 | 0.837 | 88.6 ± 1.4 | 91.1 ± 1.9 | 0.763 |
| **HbA1c (%)** | 8.25 ± 0.35 | 8.14 ± 0.54 | 0.998 | | | |
| | C-Peptide (Case) | | | C-Peptide (Control) | | |
| | Lowest Quartile (n = 28) | Highest Quartile (n = 52) | *Sig.* | Lowest Quartile (n = 35) | Highest Quartile (n = 38) | *Sig.* |
| **AIP** | 0.47 ± 0.05 | 0.64 ± 0.04 | 0.052 | 0.29 ± 0.04 | 0.56 ± 0.04 | **<0.001** |
| **CRI I** | 4.21 ± 0.19 | 4.56 ± 0.21 | 0.694 | 3.88 ± 0.13 | 4.75 ± 0.15 | **<0.001** |
| **CRI II** | 2.47 ± 0.13 | 2.41 ± 0.16 | 0.994 | 2.42 ± 0.11 | 2.91 ± 0.12 | **0.028** |
| **AC** | 3.21 ± 0.19 | 3.56 ± 0.21 | 0.694 | 2.88 ± 0.13 | 3.75 ± 0.15 | **<0.001** |
| **Age** | 48.29 ± 1.06 | 47.02 ± 0.77 | 0.761 | 49.23 ± 1.15 | 47.74 ± 1.01 | 0.688 |
| **Weight (Kg)** | 65.36 ± 2.19 | 75.40 ± 1.34 | **<0.001** | 63.97 ± 1.88 | 72.47 ± 1.33 | **0.002** |
| BMI (Kg/m$^2$) | 24.51 ± 0.66 | 27.33 ± 0.29 | **<0.001** | 23.42 ± 0.61 | 26.36 ± 0.37 | **0.001** |
| **WC (cm)** | 87.29 ± 1.75 | 95.19 ± 1.12 | **0.001** | 82.31 ± 1.75 | 93.87 ± 1.09 | **<0.001** |
| **WHR** | 0.92 ± 0.01 | 0.93 ± 0.01 | 0.658 | 0.86 ± 0.01 | 0.93 ± 0.01 | **<0.001** |
| **SBP (mmHg)** | 113.6 ± 2.3 | 111.9 ± 1.6 | 0.940 | 108.7 ± 2.1 | 111.0 ± 2.3 | 0.868 |
| **DBP (mmHg)** | 73.4 ± 1.7 | 74.7 ± 1.1 | 0.912 | 69.9 ± 1.9 | 72.5 ± 1.8 | 0.697 |
| **FBG (mg/dl)** | 205.8 ± 15.9 | 164.0 ± 7.3 | **0.042** | 87.3 ± 2.0 | 92.3 ± 1.6 | 0.204 |
| **HbA1c (%)** | 9.04 ± 0.62 | 7.97 ± 0.40 | 0.354 | | | |

* Independent t-test analyzed the differences (Mean ± SE) lipoprotein (a) and C-Peptide levels that dichotomized based on median of atherogenic indices of plasma for case and control groups.

¶ Atherogenic Index of Plasma (AIP); Castelli Risk Index I (CRI I); Castelli Risk Index II (CRI II); Atherogenic Coefficient (AC).

‡ Body Mass Index (BMI); Waist Circumference (WC); Waist to Hip Ratio (WHR); Systolic Blood Pressure (SBP); Diastolic Blood Pressure (DBP); Fasting Blood Glucose (FBG); Glycosylated Hemoglobin (HbA1c).

only in control male participants with the highest dichotomies of AIP and CRI I. Generally, the levels of serum C-peptide in the case and control males and females increased with the highest dichotomies of AIP and CRI I (Table 2).

Results showed that AIP levels significantly decreased ($P = 0.001$) in cases with the highest quartile of Lp(a). The levels of CRI II and SBP significantly increased ($P = 0.035$) and decreased ($P = 0.019$) in controls with the highest quartile of Lp(a), respectively. The weight, BMI, and WC significantly increased ($P<0.001$, $P<0.001$, and $P = 0.001$, respectively) in the cases with the highest quartile of C-Peptide. The AIP, CRI I, CRI II, AC, weight, BMI, WC, and WHR significantly increased ($P<0.001$, $P<0.001$, $P = 0.028$, $P<0.001$, $P = 0.002$, $P = 0.001$, $P<0.001$, and $P<0.001$, respectively) in controls with the highest quartile of C-peptide (Table 3).

**Table 4. Mean ± SE\* of atherogenic indices¶ of plasma according to lowest and highest dichotomies of lipoprotein (a) and C-Peptide.**

| Atherogenic Indices | Lipoprotein (a) (Case-Male) | | | Lipoprotein (a) (Control-Male) | | |
|---|---|---|---|---|---|---|
| | Lowest Dichotomy (n = 45) | Highest Dichotomy (n = 25) | Sig. | Lowest Dichotomy (n = 31) | Highest Dichotomy (n = 36) | Sig. |
| AIP | 0.62 ± 0.05 | 0.44 ± 0.03 | **0.002** | 0.46 ± 0.05 | 0.41 ± 0.04 | 0.436 |
| CRI I | 4.59 ± 0.20 | 4.17 ± 0.19 | 0.172 | 4.17 ± 0.16 | 4.54 ± 0.16 | 0.104 |
| CRI II | 2.44 ± 0.12 | 2.57 ± 0.18 | 0.524 | 2.48 ± 0.12 | 2.96 ± 0.13 | **0.008** |
| | Lipoprotein (a) (Case-Female) | | | Lipoprotein (a) (Control-Female) | | |
| | Lowest Dichotomy (n = 30) | Highest Dichotomy (n = 25) | Sig. | Lowest Dichotomy (n = 33) | Highest Dichotomy (n = 27) | Sig. |
| AIP | 0.68 ± 0.06 | 0.43 ± 0.04 | **0.001** | 0.34 ± 0.05 | 0.36 ± 0.04 | 0.796 |
| CRI I | 4.94 ± 0.34 | 3.91 ± 0.18 | **0.010** | 3.98 ± 0.16 | 4.37 ± 0.19 | 0.125 |
| CRI II | 2.69 ± 0.25 | 2.30 ± 0.17 | 0.214 | 2.44 ± 0.13 | 2.84 ± 0.16 | 0.059 |
| | C-Peptide (Case-Male) | | | C-Peptide (Control-Male) | | |
| | Lowest Dichotomy (n = 32) | Highest Dichotomy (n = 37) | Sig. | Lowest Dichotomy (n = 33) | Highest Dichotomy (n = 34) | Sig. |
| AIP | 0.50 ± 0.04 | 0.60 ± 0.05 | 0.129 | 0.34 ± 0.04 | 0.52 ± 0.04 | **0.001** |
| CRI I | 4.20 ± 0.16 | 4.68 ± 0.23 | 0.101 | 3.96 ± 0.12 | 4.76 ± 0.17 | **<0.001** |
| CRI II | 2.47 ± 0.13 | 2.53 ± 0.16 | 0.760 | 2.46 ± 0.11 | 3.01 ± 0.13 | **0.002** |
| | C-Peptide (Case-Female) | | | C-Peptide (Control-Female) | | |
| | Lowest Dichotomy (n = 20) | Highest Dichotomy (n = 35) | Sig. | Lowest Dichotomy (n = 31) | Highest Dichotomy (n = 30) | Sig. |
| AIP | 0.45 ± 0.06 | 0.64 ± 0.05 | **0.023** | 0.27 ± 0.04 | 0.44 ± 0.04 | **0.005** |
| CRI I | 4.23 ± 0.26 | 4.61 ± 0.30 | 0.401 | 3.89 ± 0.18 | 4.40 ± 0.16 | **0.043** |
| CRI II | 2.49 ± 0.17 | 2.51 ± 0.22 | 0.963 | 2.47 ± 0.16 | 2.75 ± 0.13 | 0.183 |

\* Independent t-test analyzed the differences (Mean ± SE) lipoprotein (a) and C-Peptide levels that dichotomized based on median of atherogenic indices of plasma for case and control groups.

¶ Atherogenic Index of Plasma (AIP); Castelli Risk Index I (CRI I); Castelli Risk Index II (CRI II).

Results (Table 4) indicated that the AIP levels in the case-male participants and AIP and CRI I in the case-female participants significantly decreased ($P = 0.002$, $P = 0.001$, and $P = 0.010$, respectively) with the highest dichotomy of Lp(a). However, the CRI II levels in control-male significantly increased ($P = 0.008$) with the highest dichotomy of Lp(a). The AIP levels in the case-female group significantly increased ($P = 0.023$) with the highest dichotomy of C-peptide. The levels of plasma AIP, CRI I, and CRI II in the control-male group ($P = 0.001$, $P < 0.001$, and $P = 0.002$, respectively); and the levels of plasma AIP and CRI I in the control-female group ($P = 0.005$ and $P = 0.043$, respectively) significantly increased with the highest dichotomies of C-peptide (Table 4).

Table 5 revealed that there were the negative weak significant relationships between Lp(a) with AIP (-0.309) in case-male and with AIP and CRI I (-0.303 and -0.271, respectively) in case-female. The weak significant relationships were found between C-peptide with AIP in case-male and case-female (0.246 and 0.296, respectively). However, a moderately significant relationship was found between C-peptide and AIP (0.432) in control-male and weak significant relationships were found between C-peptide with CRI I and CRI II (0.370 and 0.276, respectively) in control-male and with AIP (0.308) in control-female (Table 5).

Results (Table 6) indicated that the weight, BMI, and WC ($P < 0.001$, $P = 0.001$ and $P < 0.001$, respectively) in the case-male group and the weight, BMI, WC, and WHR ($P < 0.001$, $P < 0.001$, $P < 0.001$, and $P = 0.021$, respectively) in the control-male group significantly increased with the highest dichotomy of C-peptide. The WC ($P = 0.039$) in the case-female group and WC and WHR ($P = 0.006$ and $P = 0.003$, respectively) in the control-female group significantly increased with the highest dichotomy of C-peptide.

**Table 5. Pearson correlation coefficients for the relationship between lipoprotein (a) and C-Peptide with atherogenic indices of plasma according to gender.**

| Atherogenic Indices | Atherogenic Index of Plasma | Castelli Risk Index I | Castelli Risk Index II | Atherogenic Index of Plasma | Castelli Risk Index I | Castelli Risk Index II |
|---|---|---|---|---|---|---|
| | Case-Male | | | Case-Female | | |
| Lipoprotein | -0.309 | -0.120 | 0.137 | -0.330 | -0.271 | -0.161 |
| (a) | **0.009** | 0.322 | 0.268 | **0.014** | **0.045** | 0.253 |
| C-Peptide | 0.246 | 0.224 | 0.047 | 0.296 | 0.088 | -0.011 |
| | **0.042** | 0.064 | 0.707 | **0.028** | 0.522 | 0.937 |
| | Control-Male | | | Control-Female | | |
| Lipoprotein | -0.126 | 0.093 | 0.218 | -0.100 | 0.035 | 0.117 |
| (a) | 0.310 | 0.452 | 0.077 | 0.447 | 0.791 | 0.374 |
| C-Peptide | 0.432 | 0.370 | 0.276 | 0.308 | 0.159 | 0.077 |
| | **0.002** | **0.002** | **0.024** | **0.016** | 0.220 | 0.557 |

**Table 6. Mean ± SE* of cardiometabolic biomarkers‡ of plasma according to dichotomizing of lipoprotein (a) and C-Peptide levels.**

| Atherogenic Indices | Lipoprotein (a) (Case-Male) | | | Lipoprotein (a) (Control-Male) | | |
|---|---|---|---|---|---|---|
| | Lowest Dichotomy (n = 45) | Highest Dichotomy (n = 25) | Sig. | Lowest Dichotomy (n = 31) | Highest Dichotomy (n = 36) | Sig. |
| Age | 49.42 ± 0.78 | 48.28 ± 1.11 | 0.395 | 51.42 ± 0.88 | 48.56 ± 0.98 | **0.035** |
| Weight (Kg) | 74.53 ± 1.50 | 73.80 ± 2.16 | 0.777 | 71.35 ± 1.73 | 71.42 ± 1.86 | 0.981 |
| BMI (Kg/m$^2$) | 26.08 ± 0.38 | 25.21 ± 0.58 | 0.198 | 24.72 ± 0.58 | 24.25 ± 0.57 | 0.570 |
| WC (cm) | 94.38 ± 1.08 | 88.67 ± 1.57 | 0.453 | 89.29 ± 1.59 | 91.28 ± 1.58 | 0.381 |
| WHR | 0.94 ± 0.01 | 0.95 ± 0.01 | 0.377 | 0.91 ± 0.01 | 0.93 ± 0.01 | 0.184 |
| SBP (mmHg) | 112.3 ± 2.0 | 114.7 ± 2.4 | 0.460 | 114.5 ± 2.3 | 112.4 ± 1.6 | 0.435 |
| DBP (mmHg) | 74.7 ± 1.3 | 72.8 ± 1.7 | 0.379 | 74.4 ± 1.7 | 73.3 ± 1.6 | 0.664 |
| FBG (mg/dl) | 175.4 ± 10.9 | 181.2 ± 13.7 | 0.745 | 90.4 ± 1.8 | 90.3 ± 1.7 | 0.964 |
| HbA1c (%) | 8.50 ± 2.17 | 8.59 ± 2.43 | 0.904 | | | |
| | Lipoprotein (a) (Case-Female) | | | Lipoprotein (a) (Control-Female) | | |
| | Lowest Dichotomy (n = 30) | Highest Dichotomy (n = 25) | Sig. | Lowest Dichotomy (n = 33) | Highest Dichotomy (n = 27) | Sig. |
| Age | 46.30 ± 1.00 | 44.88 ± 0.94 | 0.312 | 45.84 ± 0.98 | 45.84 ± 0.98 | 0.940 |
| Weight (Kg) | 66.22 ± 1.38 | 65.56 ± 2.22 | 0.796 | 62.82 ± 1.68 | 66.33 ± 1.70 | 0.150 |
| BMI (Kg/m$^2$) | 26.84 ± 0.44 | 26.16 ± 0.62 | 0.364 | 25.90 ± 0.64 | 25.82 ± 0.59 | 0.924 |
| WC (cm) | 88.67 ± 1.57 | 87.64 ± 2.26 | 0.704 | 86.63 ± 2.13 | 87.33 ± 1.89 | 0.812 |
| WHR | 0.91 ± 0.01 | 0.89 ± 0.01 | 0.295 | 0.87 ± 0.02 | 0.87 ± 0.02 | 0.861 |
| SBP (mmHg) | 112.8 ± 2.4 | 111.1 ± 2.3 | 0.600 | 111.4 ± 2.3 | 101.3 ± 2.5 | **0.005** |
| DBP (mmHg) | 75.6 ± 1.3 | 71.9 ± 2.0 | 0.098 | 71.8 ± 1.7 | 65.9 ± 2.2 | **0.035** |
| FBG (mg/dl) | 183.7 ± 9.7 | 188.9 ± 15.6 | 0.780 | 88.3 ± 1.5 | 91.9 ± 2.8 | 0.238 |
| HbA1c (%) | 8.08 ± 2.12 | 8.26 ± 2.62 | 0.811 | | | |
| | C-Peptide (Case-Male) | | | C-Peptide (Control-Male) | | |
| | Lowest Dichotomy (n = 32) | Highest Dichotomy (n = 37) | Sig. | Lowest Dichotomy (n = 33) | Highest Dichotomy (n = 34) | Sig. |
| Age | 48.84 ± 0.99 | 49.41 ± 0.83 | 0.662 | 50.91 ± 1.00 | 48.88 ± 0.92 | 0.139 |
| Weight (Kg) | 68.94 ± 1.75 | 78.81 ± 1.38 | **<0.001** | 66.82 ± 1.79 | 75.82 ± 1.46 | **<0.001** |
| BMI (Kg/m$^2$) | 24.51 ± 0.53 | 26.79 ± 0.31 | **0.001** | 23.06 ± 0.62 | 25.84 ± 0.41 | **<0.001** |
| WC (cm) | 90.28 ± 1.33 | 97.03 ± 1.01 | **<0.001** | 85.42 ± 1.59 | 95.15 ± 1.08 | **<0.001** |
| WHR | 0.94 ± 0.01 | 0.95 ± 0.01 | 0.161 | 0.90 ± 0.01 | 0.93 ± 0.01 | **0.021** |
| SBP (mmHg) | 113.0 ± 2.3 | 113.0 ± 2.1 | 0.995 | 112.4 ± 2.0 | 114.3 ± 1.9 | 0.505 |
| DBP (mmHg) | 74.0 ± 1.7 | 73.9 ± 1.3 | 0.975 | 71.7 ± 1.8 | 75.9 ± 1.4 | 0.157 |
| FBG (mg/dl) | 191.8 ± 14.1 | 166.3 ± 10.1 | 0.139 | 88.6 ± 1.8 | 92.1 ± 1.6 | 0.072 |
| HbA1c (%) | 8.84 ± 0.46 | 8.22 ± 0.50 | 0.369 | | | |

*(Continued)*

**Table 6.** (Continued)

| Atherogenic Indices | Lipoprotein (a) (Case-Male) | | | Lipoprotein (a) (Control-Male) | | |
|---|---|---|---|---|---|---|
| | Lowest Dichotomy (n = 45) | Highest Dichotomy (n = 25) | *Sig.* | Lowest Dichotomy (n = 31) | Highest Dichotomy (n = 36) | *Sig.* |
| | C-Peptide (Case-Female) | | | C-Peptide (Control-Female) | | |
| | Lowest Dichotomy (n = 20) | Highest Dichotomy (n = 35) | *Sig.* | Lowest Dichotomy (n = 31) | Highest Dichotomy (n = 30) | *Sig.* |
| Age | 46.60 ± 1.06 | 45.11 ± 0.90 | 0.306 | 46.90 ± 0.89 | 45.10 ± 1.05 | 0.198 |
| Weight (Kg) | 63.03 ± 1.72 | 67.57 ± 1.65 | 0.079 | 63.45 ± 1.89 | 65.57 ± 1.49 | 0.381 |
| BMI (Kg/m$^2$) | 26.19 ± 0.67 | 26.73 ± 0.44 | 0.486 | 25.30 ± 0.63 | 26.46 ± 0.57 | 0.177 |
| WC (cm) | 84.60 ± 1.94 | 90.26 ± 1.69 | **0.039** | 83.10 ± 2.03 | 90.70 ± 1.74 | **0.006** |
| WHR | 0.88 ± 0.01 | 0.91 ± 0.01 | 0.111 | 0.84 ± 0.01 | 0.90 ± 0.02 | **0.003** |
| SBP (mmHg) | 113.1 ± 2.7 | 111.4 ± 2.1 | 0.629 | 104.8 ± 1.9 | 109.3 ± 3.0 | 0.215 |
| DBP (mmHg) | 72.8 ± 1.8 | 74.6 ± 1.4 | 0.434 | 68.5 ± 1.8 | 70.2 ± 2.1 | 0.564 |
| FBG (mg/dl) | 220.0 ± 18.4 | 166.7 ± 7.4 | **0.013** | 88.6 ± 2.1 | 90.9 ± 2.1 | 0.428 |
| HbA1c (%) | 9.03 ± 0.68 | 7.55 ± 0.37 | 0.067 | | | |

* Independent t-test analyzed the differences (Mean ± SE) lipoprotein (a) and C-Peptide levels that dichotomized based on median of atherogenic indices of plasma for case and control groups.

‡ Body Mass Index (BMI); Waist Circumference (WC); Waist to Hip Ratio (WHR); Systolic Blood Pressure (SBP); Diastolic Blood Pressure (DBP); Fasting Blood Glucose (FBG); Glycosylated Hemoglobin (HbA1c).

Figs 1 and 2 exhibited the correlation of Lp(a) and C-peptide with atherogenic indices of plasma according to both genders, respectively.

## Discussion

We sought the association between serum Lp(a) and C-peptide levels as two predictors with atherogenic indices of plasma as novel predictive biomarkers and cardiometabolic biomarkers in patients with T2DM based on genders in KERCADR cohort study as an Iranian community. Our principal purpose was to find out the alignment of the role of these two predictors together with atherogenic indices of plasma and other cardiometabolic biomarkers in predicting the occurrence of cardiovascular disease in patients with T2DM and compare it with control participants. These biomarkers are being applied for assessing cardiovascular risk.

We revealed that serum C-peptide levels significantly increased among case females than control females. Although, serum C-peptide levels in case and control male and female participants increased with the highest dichotomies of AIP and CRI I, the difference between participants with the lowest and the highest dichotomies among control males was significant. Accordingly, plasma AIP levels in case females significantly increased with the highest dichotomy of C-peptide. Plasma AIP, CRI I, and CRI II levels in control males and plasma AIP and CRI I levels in control females significantly increased with the highest dichotomy of C-peptide. The correlation coefficient analysis proved these results. Hence, there was a significant positive correlation between C-peptide with AIP, CRI I, and CRI II; and C-peptide with AIP among control male and female participants, respectively. Therefore, according to the results of other studies, it revealed that increasing this biomarker in healthy participants can be a potential risk for premature coronary events without T2DM. Raised serum C-peptide as a predictor of premature coronary incidents than impaired glucose tolerance is associated with the pathogenesis of atherosclerosis and the occurrence of MI and CAD in the popular community [2, 7]. Hence, another study demonstrated that raised C-peptide levels significantly increased the risk of cardiovascular mortality independently of recognized diabetes in patients with atherosclerotic cardiovascular disease (ASCVD) [8]. Alternatively, we showed that plasma AIP levels in case

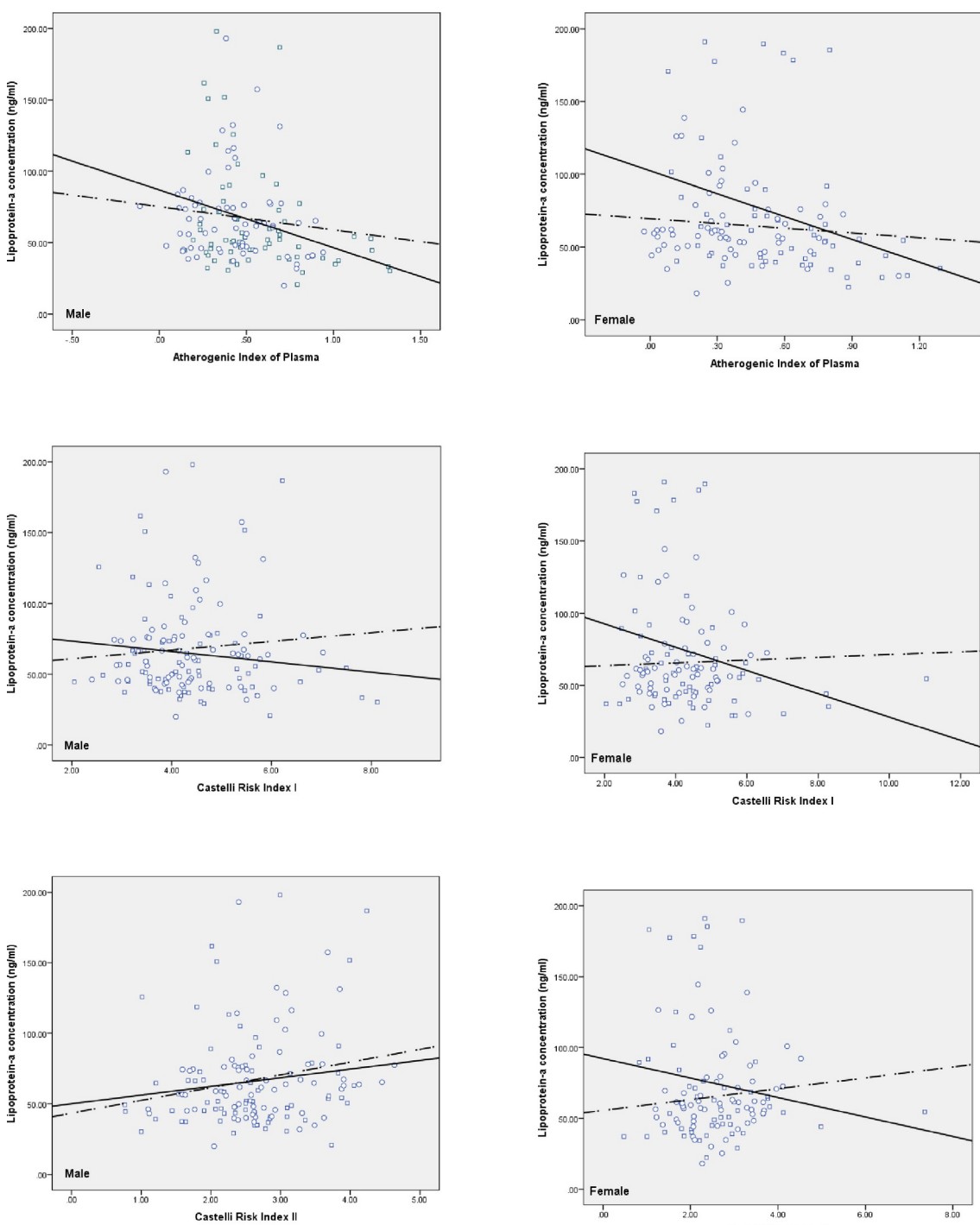

**Fig 1. Correlation\* of Lipoprotein (a) with Atherogenic Indices of Plasmaꞁ according to both genders.** Control: circle; Case: square; Control regression line: dashed line; Case regression line: solid line. Atherogenic Index of Plasma (AIP); Castelli Risk Index I (CRI I); Castelli Risk Index II (CRI II).

females than case males significantly increased with the highest dichotomy of C-Peptide. The gender difference might occur in many studies. In one study, the relative risks of T2DM, dyslipidemia, and hypertension for women were higher than for men in the incidence of MI [21].

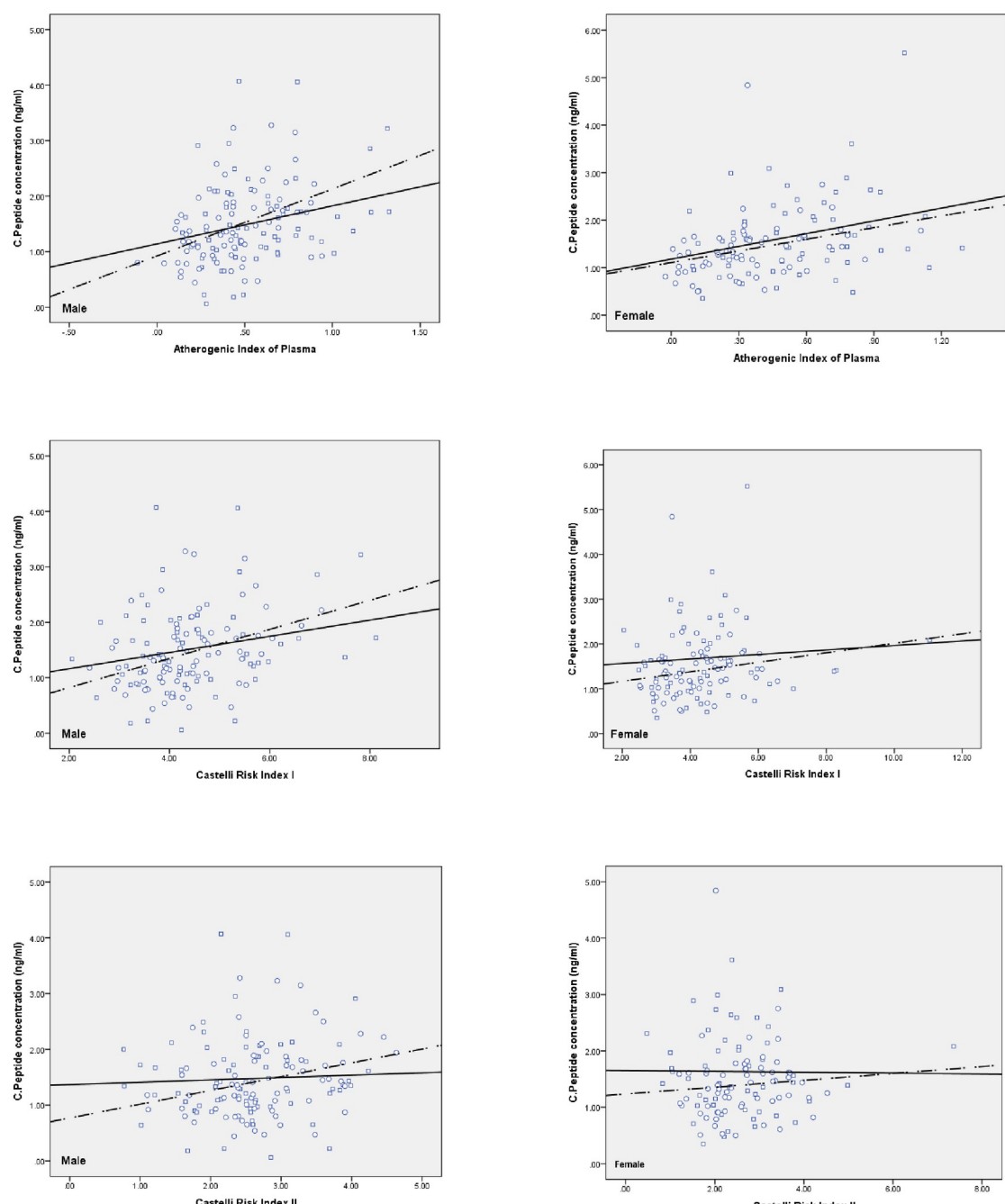

**Fig 2. Correlation of C-Peptide with atherogenic indices of plasmaç according to both genders.** Control: circle; Case: square; Control regression line: dashed line; Case regression line: solid line. Atherogenic Index of Plasma (AIP); Castelli Risk Index I (CRI I); Castelli Risk Index II (CRI II).

On the other hand, raised serum C-peptide with traditional risk factors such as obesity and dyslipidemia augmented the chance of the progress of atherosclerosis and CAD in patients with T2DM and without T2DM [4, 6]. Consistent with the other studies, in the current study, the weight, BMI, and WC significantly increased in case participants with the highest quartile of C-Peptide. However, the AIP, CRI I, CRI II, AC, weight, BMI, WC, and WHR significantly

increased in control participants with the highest quartile of C-Peptide. Based on a fundamental perspective, we revealed that case and control males with general and visceral obesity are at risk of increased C-peptide. However, case and control females with visceral obesity were at risk for increased C-peptide. Hence, general and visceral obesity and increased atherogenic indices of plasma were associated with increased C-peptide in the control participants. Plasma AIP levels in case females than case males significantly increased with the highest dichotomy of C-Peptide. Therefore, C-peptide might influence the worse prognosis of atherosclerosis and MI in the general population without T2DM. Other studies confirmed this claim. Raised serum C-peptide levels were associated with all-cause and cardiovascular mortality among individuals without T2DM [22] and patients who underwent coronary angiography [23]. Therefore, raised serum C-peptide levels than the other glycemic indices were a risk predictor of cardiovascular and CVD death among participants without T2DM [24]. Consequently, it suggested monitoring the serum C-peptide in seemingly healthy individuals with obesity and cardiometabolic risk markers.

In general, we found that atherogenic indices of plasma remarkably reduced in the highest dichotomy of Lp(a) in case participants. Consequently, no significant increases were demonstrated for atherogenic indices of plasma and the other cardiometabolic biomarkers between the highest and the lowest Lp(a) for case participants. However, these indices remarkably increased in the highest dichotomy of Lp(a) in control participants. Regardless of gender, AIP levels significantly decreased in case participants and CRI II levels significantly increased in control participants with the highest quartile of Lp(a).

Accordingly, two fundamental opinions become noticeable. The first opinion was that there was a direct relationship between Lp(a) levels with TC and LDL-C and a reverse relationship between Lp(a) levels with HDL-C in healthy participants. The results from a research indicated that serum Lp(a) levels have a positive correlation with TC and LDL-C in patients with T2DM [16]. These results also revealed that serum Lp(a) levels had no significant associations with glycemic indices, HDL-C, BMI, and high blood pressure [16].

The second opinion, these consequences indicated that Lp(a) might be the independent risk biomarker for T2DM in the current study. It was proven that raised Lp(a) was an independent predictable risk biomarker for the severity of new-onset CAD in patients with T2DM [10, 11]. Hence, Elevated Lp(a) and raised LDL-C were independent prognosticators of the intensity of CAD [12]. Raised Lp(a) levels might affect the prognosis in patients with pre-DM with stable CAD [13]. In addition to raised Lp(a) concentrations that were associated with the severity of CAD in T2DM, patients with MI and acute coronary syndrome had higher Lp(a) concentrations [25].

Two approaches were established for Lp(a) levels in patients with T2DM in the current study. The first approach was that nutritional and pharmacological interventions and lifestyle changes in patients with T2DM might result in decreased serum Lp(a) levels. Chen and colleagues indicated that raised Lp(a) levels were also an independent risk biomarker in patients with major adverse cardiovascular events (MACEs). It appeared that statin therapy was a protective and favorable approach for patients with MACEs [26]. Findings from a meta-analysis revealed that a reduced LDL-C was observed after the beginning of statin therapy without a significant change in Lp(a) levels [27]. The results of another meta-analysis proposed that ezetimibe therapy either alone or incorporate with a statin does not modify plasma Lp(a) levels [28]. On the other hand, fibrates have a remarkably pronounced effect in alleviating plasma Lp (a) levels than statins. Integration of fibrates to statins can intensify the Lp(a)-lowering effect of statins [29]. In clinical guidelines, these drugs are recommended for the control and amelioration of dyslipidemia in patients with diabetes or hypercholesterolemia. However, in spite of the extensive access of insulin and other medications in Iran for patients with diabetes, the

estimated national control of hyperglycemia, hyperlipidemia and hypertension exclusively for old women and young men remains insufficient and the prevalence of chronic cardiovascular disease among these patients are relatively high in Iran [30]. Nevertheless, a Report of the American College of Cardiology claimed that clinical guidelines dispense advices applicable to patients with or at risk of developing CVD. The major aim of these guidelines to ameliorate quality of care and support patients' interests. Guidelines should not substitute clinical judgment. These guidelines are efficient only when complied both patients and clinicians [31].

The second approach was whether a reduction in Lp(a) levels reduces the risk of the severity of CAD in patients with T2DM? Undoubtedly, we could confidently say that reducing Lp(a) levels is crucial in controlling and reducing the severity of CAD in these patients, however without improving the other cardiometabolic biomarker, the reduction of raised Lp(a) levels may be a little misleading. Tsimikas claimed that some studies had mentioned that patients with low serum Lp(a) levels were associated with a higher risk of occurrence T2DM. On the other hand, patients with raised serum Lp(a) levels had lower exposure to risk [32].

The reality of Lp(a) is that the levels of this predictable risk biomarker should be minimized in patients with T2DM and seemingly healthy individuals with nutritional and pharmacological interventions. Reducing raised Lp(a) levels and strict control of LDL-C would be the best approach to prevent CAD consequences. Therefore, according to recent studies, it is best to reduce raised Lp(a) levels to less than 30 [26]. The effectiveness is better on-statin therapy, especially in younger individuals [27]. Results from a biracial cohort revealed that raised Lp(a) levels in Caucasian patients with prediabetes or T2DM were associated with further augmented ASCVD risk [33]. Therefore, individuals with the higher Lp(a) levels had a higher occurrence of complications such as macrovascular disease and calcified aortic valve disease than patients with very low Lp(a) levels [34].

## Strengths and limitation

There are several strengths in the current study. In this nested case-control study, the participants were selected from the second phase of the KERCADR cohort study as an Iranian community. Participants with diabetes and control were randomly selected and matched based on inclusion criteria. High blood pressure, BMI $\geq$ 30, and known potential confounders were exclusion criteria for enrollment. All possible analyzes were performed between participants with diabetes and controls by gender. One of the limitations of the current study was lack of selection some participants with diabetes who had the other risk factors as specified and impressive confounding variables such as having history or high blood pressure and BMI$\geq$30 and did not enroll in our investigation.

## Conclusion

We concluded that there was a significant association between the highest quartile of C-peptide and higher anthropometric values in case participants and higher atherogenic indices of plasma and anthropometric values in control participants. Therefore, Case and control males with general and visceral obesity had exposure to increased C-peptide. However, case and control females with visceral obesity are exposure to increased C-peptide. Hence, general and visceral obesity and increased atherogenic indices of plasma were associated with increased C-peptide in the seemingly healthy participants. Raised serum C-peptide than raised Lp(a) levels may be a prior predictor for cardiometabolic disease risk in healthy participants and patients with T2DM with increased cardiometabolic biomarkers. Therefore, C-peptide may influence the worse prognosis of atherosclerosis and MI in the general population without T2DM. Consequently, it suggested monitoring the serum C-peptide in seemingly healthy individuals with

obesity and cardiometabolic risk markers. Raised Lp(a) levels may be a risk independent biomarker for T2DM in the current study. Reducing raised Lp(a) levels to less than 30ng/ml with strict control of LDL-C would be the best approach to prevent CAD consequences. Finally, it is suggested that a screening system be set up to measure the Lp(a) levels in the community for seemingly healthy people or individuals with one or more risk cardiometabolic biomarkers.

## Supporting information

**S1 File.**
(RAR)

## Acknowledgments

The authors are also grateful to participants that contributed in this study. We special thank Dr. Yaser Masoumi-Ardakani for measuring and determining C-peptide and Lp(a) levels.

## Author Contributions

**Conceptualization:** Mohammad Reza Mahmoodi.

**Data curation:** Hamid Najafipour.

**Formal analysis:** Mohammad Reza Mahmoodi.

**Funding acquisition:** Mohammad Reza Mahmoodi.

**Investigation:** Mohammad Reza Mahmoodi.

**Methodology:** Mohammad Reza Mahmoodi.

**Project administration:** Hamid Najafipour.

**Resources:** Hamid Najafipour.

**Supervision:** Hamid Najafipour.

**Validation:** Hamid Najafipour.

**Writing – original draft:** Mohammad Reza Mahmoodi, Hamid Najafipour.

**Writing – review & editing:** Mohammad Reza Mahmoodi.

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
