## [Decision Letter · Decision Letter 0]

15 Mar 2022

PONE-D-22-01443Association of C-peptide and lipoprotein(a) as two predictors with cardiometabolic biomarkers in patients with type 2 diabetes in KERCADR population-based studyPLOS ONE

Dear Dr. Mahmoodi,

Thank you for submitting your manuscript to PLOS ONE. After careful consideration, we feel that it has merit but does not fully meet PLOS ONE’s publication criteria as it currently stands. Therefore, we invite you to submit a revised version of the manuscript that addresses the points raised during the review process.

We look forward to receiving your revised manuscript.

Kind regards,

Kanhaiya Singh, Ph.D

Academic Editor

PLOS ONE

Journal Requirements:

(The authors are grateful to Vice chancellor for Research of Kerman University of Medical Sciences in this study. The authors are also grateful to participants that contributed in this study. We special thank Dr. Yaser Masoumi-Ardakani for measuring and determining C-peptide and Lp(a) levels. )

 (NO. The Vice chancellor for Research of Kerman University of Medical Sciences had no role in study design, data collection and analysis, decision to publish, or preparation of the manuscript)

Additional Editor Comments:

Although the reviewers have found this study interesting, they have raised some concern about the significance and novelty of this study. Please also elaborate about the guidelines being used to treat patients with T2DM in your population.

Reviewers' comments:

Reviewer's Responses to Questions

**Comments to the Author**

1. Is the manuscript technically sound, and do the data support the conclusions?

Reviewer #1: Yes

Reviewer #2: Yes

2. Has the statistical analysis been performed appropriately and rigorously? 

Reviewer #1: I Don't Know

Reviewer #2: Yes

3. Have the authors made all data underlying the findings in their manuscript fully available?

Reviewer #1: Yes

Reviewer #2: Yes

4. Is the manuscript presented in an intelligible fashion and written in standard English?

Reviewer #1: No

Reviewer #2: Yes

5. Review Comments to the Author

Reviewer #1: Ref: PONE-D-22-01443

In the present article entitled “Association of C-peptide and lipoprotein(a) as two predictors with cardiometabolic

biomarkers in patients with type 2 diabetes in KERCADR population-based study” authors have explored the association between serum Lp(a) and C-Peptide levels as two predictors with cardiometabolic biomarkers in patients with type 2 diabetes mellitus (T2DM). The study design is simple and the results are well supported by the data. However, some points need to be addressed to make the study robust for the publication.

Major concerns:

1. Giving clinical perspective to the paper: Existing medicines already are know to lower C-peptides and lipoprotein levels, so the study does little in terms of adding to the solution for these Laboratory test. Kindly mention what are the current guidelines followed in Middle East to treat T2DM

2. Methodology: Addressing the Latent period of pre-diabetes: consider giving latest references of how this is affecting the biomarker levels

2. Novelty concerns: existing biomarker for management of Type 2 DM, that is C-peptide level is well controlled with injectable Insulin and SGLT2 inhibitors are a class of prescription medicines that are FDA-approved for use with diet and exercise to lower blood sugar in adults with type 2 diabetes. Medicines in the SGLT2 inhibitor class include canagliflozin, dapagliflozin, and empagliflozin. Please mention these in the manuscript

Minor concerns:

1. Abbreviations need to be defined at the start of abstract: line 28: Lp(A),

Line 34: AIP, CRI

2. Fig 1,2: AIP, CRI, CRII

legends do not represent the abbreviations properly

Best regards,

Reviewer #2: In this manuscript the authors are looking at the role of Lp(a) and C-peptide levels as biomarkers for assessing cardiovascular risk. The study is being performed with both healthy and T2DM patients to understand if there is any association between these biomarkers along with other atherogenic indices of plasma for predicting the risk of cardiovascular diseases.

Overall, the manuscript is well drafted and the methods used are appropriate. The findings are well supported by the data. There are a few minor suggestions that I have highlighted below.

1. The authors mention that the data is fully available without restriction, but also mention that restrictions apply due to licensing issues. Please clarify this point.

2. Line 101-102, please briefly explain the study design here. Interested readers can look up the citations, however, a brief explanation here can help the readers move along this manuscript.

3. Line 115-116, for the T2DM eligibility criteria, the authors mention ‘6) participants with T2DM receive either therapeutic diet or therapeutic diet with a combination of oral anti glycemic drugs’. Was this only for selecting participants, or were the study participants given this diet? I did not see this data point being used anywhere in the remaining manuscript, hence, the question.

4. Materials and methods, for ‘Determination of serum lipoprotein (a)’, ELISA has been explained in detail, whereas, for ‘Determination of serum C-peptide’, that is not the case. Given the order of occurrence, details should be mentioned for serum C-peptide and in the case of Lp(a), authors can reference the C-peptide assay.

5. Line 376-377, the limitations of the study are not defined adequately. Dropping out participants was due to their ineligibility for the study. How is that a limitation? Please define.

6. Line 392-393, please combine both the sentences. Also, kindly have a final concluding statement for the manuscript.

7. Line 56, should be ‘and has been revealed to display…’.

8. Line 58, should be ‘as a biomarker for…’.

9. Line 90, please define ‘CHD’.

10. Line 289, ‘on the one hand’, does not go along with the sentence. Please modify accordingly.

11. Line 383-384, should be ‘had exposure to…’.

6. PLOS authors have the option to publish the peer review history of their article (what does this mean?). If published, this will include your full peer review and any attached files.

Reviewer #1: No

Reviewer #2: No

---

## [Author Response · Author response to Decision Letter 0]

11 Apr 2022

Dear Dr. Kanhaiya Singh

Academic Editor

PLOS ONE

Thank you so much for this occasion to answer peer review and revise our manuscript (Submission PONE-D-22-01443 is revised). The following, my written answer to honorable referees’ comments and fully manuscript revision was done. Thanks for helpful comments. All comments and suggestions were corrected and are in yellow highlighted. 

Reply to the respected academic editor and honorable reviewers’ comments

Academic editor

Our manuscript (highlighted in yellow) was modified based on the PLoS ONE submission guideline.

Thank you for your comment. The individuals aged between 15 to 75 years were enrolled in the household survey by non-proportional to size one-stage cluster sampling on Kerman province residences. The written informed consent was signed by all of the participants after ensuring their well understanding of the participation in the survey. Therefore, according to the age range of the patients, in our study, the patients themselves presented informed consent.

Thank you very much for your comment. As you know, The grant number of this research was (Grant ID: 95000643), however, the Ethics Certificate code was (IR.KMU.REC. 1395.938).

4a. Please include your amended statements within your cover letter; we will change the online submission form on your behalf. 

 (NO. The Vice chancellor for Research of Kerman University of Medical Sciences had no role in study design, data collection and analysis, decision to publish, or preparation of the manuscript)

Thank you for your comment and description. 

a. I modified the Acknowledgment in the text of manuscript and deleted the Funding source from it.

b. You are right. With permission, I deleted the Declaration from the text of manuscript and amended statements to Cover letter. 

c. I removed any funding-related text from the manuscript.

I mean, the Vice chancellor for research had no role in how to buy the kits, study design, data collection, and make decisions in other cases (we hadn't any potential conflict of interest relevant to this manuscript).

d. All right. Thank you very much.

I beg your pardon for any inconvenience. You are right.

I have a final SPSS dataset from analysis of our data. There is an Excel dataset file for Lipoprotein (A) and C-peptide data that had transferred from Dr. Yaser Masoumi-Ardakani to me.

I agree with your suggestion. I can transfer those files when resubmit our manuscript. 

Although the reviewers have found this study interesting, they have raised some concern about the significance and novelty of this study. Please also elaborate about the guidelines being used to treat patients with T2DM in your population.

An updated and native version of the Diabetes Clinical Guideline consists of 150 pages in Persian has been published based on the general framework for providing services to diabetic patients in May 2021. 

The guideline covers topics related to the diagnosis, prevention and care of diabetes, nutrition, education, insulin use, oral medication, control of dyslipidemia, treatment of hypertension, use of antiplatelet in diabetes, vaccination, treatment of hypoglycemia, prevention and treatment of diabetes complications, diagnosis and treatment of Gestational Diabetes, fasting, oral diseases, and Diabetes Referral Protocol for Psychological Issues.

Only one page is in English: A diagram about glucose-lowering medication in type 2 diabetes: 2021 ADA Professional Practice Committee (PPC) adaptation of Davies et al.

Major concerns of Reviewer 1

1. Giving clinical perspective to the paper: Existing medicines already are known to lower C-peptides and lipoprotein levels, so the study does little in terms of adding to the solution for these Laboratory test. Kindly mention what are the current guidelines followed in Middle East to treat T2DM

Thank you for your comments. I described to respected academic editor about an updated and native version of the Diabetes Clinical Guideline consists of 150 pages in Persian that compiled based on Evidence-based Medicine in Diabetes mellitus and perspectives and authorities of endocrinologist/internist.

Unfortunately, this guideline hasn’t any bibliographic identification in Iran’s National Library. However, I can’t refer to it in Reference section of manuscript.

Only one page is in English: A diagram about glucose-lowering medication in type 2 diabetes: 2021 ADA Professional Practice Committee (PPC) adaptation of Davies et al.

As you know, lipid and lipoprotein-lowering agents such as Ezetimibe, Fibrates, and Statins can influence on lowering lipoprotein (a). 

I provided two new references about effectiveness medications on plasma Lp(a) levels (refs: 28 & 29 highlighted in yellow). These two new references and previous reference (27) and the other references could provide a clinical perspective.

With permission, I provided the authors’ perspective that compiled the updated and native version of diabetes clinical guideline in Persian about adherence of diabetic patients to their medications (ref. 30). 

2. Methodology: Addressing the Latent period of pre-diabetes: consider giving latest references of how this is affecting the biomarker levels

Dear referee, as you know, one of the main aim was to find out whether there was an association between serum C-peptide and Lp(a) levels as two predictors with biomarkers of cardiometabolic disease risk in patients with T2DM and healthy people based on genders in KERCADR study as an Iranian community.

Let us not discuss these biomarkers in patients with prediabetes, due to the complexity of the topics. 

3. Novelty concerns: existing biomarker for management of Type 2 DM that is C-peptide level is well controlled with injectable Insulin and SGLT2 inhibitors are a class of prescription medicines that are FDA-approved for use with diet and exercise to lower blood sugar in adults with type 2 diabetes. Medicines in the SGLT2 inhibitor class include canagliflozin, dapagliflozin, and empagliflozin. Please mention these in the manuscript

You are Right. These medications were pointed out in the updated and native version of Diabetes Clinical Guideline. Clinicians occasionally prescribe these medications. Not everyone is able to buy these medications due to inflation and sanctions.

1. Abbreviations need to be defined at the start of abstract: line 28: Lp(A),

Line 34: AIP, CRI

Thank you very much for your comment. I modified based on PLoS ONE guideline submission.

2. Fig 1,2: AIP, CRI, CRII

legends do not represent the abbreviations properly

Based on PLoS ONE guideline submission, I modified in text of manuscript and added title and legends to Figs 1 & 2.

Major concerns of Reviewer 2

1. The authors mention that the data is fully available without restriction, but also mention that restrictions apply due to licensing issues. Please clarify this point.

Thank you for your comment. The data of this research (SPSS file) is fully available without restriction. 

2. Line 101-102, please briefly explain the study design here. Interested readers can look up the citations, however, a brief explanation here can help the readers move along this manuscript.

I briefly described the protocol. Highlighted in yellow. 

3. Line 115-116, for the T2DM eligibility criteria, the authors mention ‘6) participants with T2DM receive either therapeutic diet or therapeutic diet with a combination of oral anti glycemic drugs’. Was this only for selecting participants, or were the study participants given this diet? I did not see this data point being used anywhere in the remaining manuscript, hence, the question.

Thank you for your comment. As you know and mentioned, this criterion has been the criterion for selecting patients with type 2 diabetes. We did not give them any meals. Only after blood sampling, a snack was given to all study participants, as well as their travel expenses. 

4. Materials and methods, for ‘Determination of serum lipoprotein (a)’, ELISA has been explained in detail, whereas, for ‘Determination of serum C-peptide’, that is not the case. Given the order of occurrence, details should be mentioned for serum C-peptide and in the case of Lp(a), authors can reference the C-peptide assay.

Dear referee, I shifted the subheading of lipoprotein(a) to subheading of C-peptide. 

5. Line 376-377, the limitations of the study are not defined adequately. Dropping out participants was due to their ineligibility for the study. How is that a limitation? Please define.

As you know, Each study is designed based on the characteristics of the same study. Our study has been a nested case-control study, and I have described in “Participants eligibility and study design” section in details. 

The proportion between cases and controls became 1:1 due to the precise control of confounding variables in the study. Hence, the study groups did not vary regarding specified and impressive confounders. Therefore, a matched healthy control was carefully chosen for each case from among participants in the KERCADR cohort study.

Therefore, we had to exclude many diabetic patients because the study has been a nested case-control study. I included some criteria in the inclusion criteria (eg. lack of high blood pressure, BMI lower than 30, ….). Hence, many diabetic patients had high blood pressure or BMI>30 and excluded from the study. These criteria were confounding variables for our study. 

6. Line 392-393, please combine both the sentences. Also, kindly have a final concluding statement for the manuscript.

Thank you very much. I combined two sentences.

With permission, final concluding statement was provided in framework of a suggestion. 

7. Line 56, should be ‘and has been revealed to display…’.

I modified it. The passive voice sentence converted to the active voice sentence.

8. Line 58, should be ‘as a biomarker for…’.

Thank you for your comment. I modified it.

9. Line 90, please define ‘CHD’.

Based on guideline submission, I defined CHD.

10. Line 289, ‘on the one hand’, does not go along with the sentence. Please modify accordingly.

Thank you for your comment. I modified accordingly.

11. Line 383-384, should be ‘had exposure to…’.

Thank you for your comment. I modified it.

---

## [Decision Letter · Decision Letter 1]

11 May 2022

Association of C-peptide and lipoprotein(a) as two predictors with cardiometabolic biomarkers in patients with type 2 diabetes in KERCADR population-based study

PONE-D-22-01443R1

Dear Dr. Mahmoodi,

We’re pleased to inform you that your manuscript has been judged scientifically suitable for publication and will be formally accepted for publication once it meets all outstanding technical requirements.

Kind regards,

Kanhaiya Singh, Ph.D

Academic Editor

PLOS ONE

Additional Editor Comments (optional):

Reviewers' comments:

Reviewer's Responses to Questions

**Comments to the Author**

1. If the authors have adequately addressed your comments raised in a previous round of review and you feel that this manuscript is now acceptable for publication, you may indicate that here to bypass the “Comments to the Author” section, enter your conflict of interest statement in the “Confidential to Editor” section, and submit your "Accept" recommendation.

Reviewer #1: All comments have been addressed

Reviewer #2: All comments have been addressed

2. Is the manuscript technically sound, and do the data support the conclusions?

Reviewer #1: Yes

Reviewer #2: Yes

3. Has the statistical analysis been performed appropriately and rigorously? 

Reviewer #1: N/A

Reviewer #2: Yes

4. Have the authors made all data underlying the findings in their manuscript fully available?

Reviewer #1: Yes

Reviewer #2: Yes

5. Is the manuscript presented in an intelligible fashion and written in standard English?

Reviewer #1: Yes

Reviewer #2: Yes

6. Review Comments to the Author

Reviewer #1: (No Response)

Reviewer #2: (No Response)

7. PLOS authors have the option to publish the peer review history of their article (what does this mean?). If published, this will include your full peer review and any attached files.

Reviewer #1: No

Reviewer #2: No

---

## [Editor Report · Acceptance letter]

13 May 2022

PONE-D-22-01443R1 

Association of C-peptide and lipoprotein(a) as two predictors with cardiometabolic biomarkers in patients with type 2 diabetes in KERCADR population-based study 

Dear Dr. Mahmoodi:

I'm pleased to inform you that your manuscript has been deemed suitable for publication in PLOS ONE. Congratulations! Your manuscript is now with our production department. 

Kind regards, 

on behalf of

Dr. Kanhaiya Singh 

Academic Editor

PLOS ONE